# A Modular Chain Bioreactor Design for Fungal Productions

**DOI:** 10.3390/biomimetics7040179

**Published:** 2022-10-27

**Authors:** Onur Kırdök, Berker Çetintaş, Asena Atay, İrem Kale, Tutku Didem Akyol Altun, Elif Esin Hameş

**Affiliations:** 1Department of Biotechnology, Graduate School of Natural and Applied Sciences, Ege University, 35040 İzmir, Türkiye; 2Department of Bioengineering, Graduate School of Natural and Applied Sciences, Ege University, 35040 İzmir, Türkiye; 3Department of Architecture, Faculty of Architecture, Dokuz Eylül University, 35390 İzmir, Türkiye; 4Department of Bioengineering, Faculty of Engineering, Ege University, 35040 İzmir, Türkiye

**Keywords:** modular chain bioreactor, solid-state fermentation, mycelium production, *Ganoderma lucidum*

## Abstract

Plastic bag bioreactors are single-use bioreactors, frequently used in solid culture fermentation. This study developed plastic bag bioreactors with more effective aeration conditions and particular connection elements that yield sensors, environmental control, and modular connectivity. This bioreactor system integrates the bags in a chain that circulates air and moisture through filtered connections. Within the present scope, this study also aimed to reveal that cultures in different plastic bags can be produced without affecting each other. In this direction, biomass production in the modular chain bioreactor (MCB) system developed in this study was compared to traditional bag systems. In addition, contamination experiments were carried out between the bags in the system, and it was observed that the filters in the developed system did not affect the microorganisms in different bags.

## 1. Introduction

Today, mycelium growing plays an important role in mushroom production and various biotechnological studies. In recent years, bag containers have been investigated as promising tools for mycelium development, going beyond traditional tray-type bioreactors [1]. According to research, although a limited gas exchange and a limited heat transfer are observed in bag systems, this provides an ideal environment for mycelium production due to low moisture loss [2]. Since the beginning of the use of bag containers, studies have been carried out on edible mushrooms, such as *Ganoderma lucidum* [3,4], *Pleurotus ostreatus* [5,6], *Trametes versicolor* [7,8], and *Agaricus bisporus* [9,10]. As a result of these studies, bag systems are used for mushroom growing in many countries for academic and commercial purposes [11]. Such systems can be divided into two groups depending on the sterility. Commercial bag cultivator systems mostly use plastic bags with needle holes, or even straw sacks, that are often vulnerable to contamination risks. More recently, micro-holed plastic bags have become popular. However, the risk of contamination continues due to the humidity and perspiration on the plastic surface. Therefore, mushroom farmers mostly prefer pasteurizing the substrate before inoculation to ensure that the fungus is the dominant organism in production. On the other hand, in academic studies, sterile conditions are mainly created to provide controlled conditions, and cultivation is carried out with the selected pure microorganism. To achieve that, bags to be used in such systems should be able to be effectively sterilized. In most cases, the preferred sterilization method is high-pressure steam heated using an autoclave. These bags must be made of a high-temperature-resistant material, such as polypropylene, to be autoclaved. The sterilization of the substrate usually requires a temperature of 121 °C, and the bag used in the system must also be resistant to high temperatures.

Bag systems are also used for mushroom cultivation and microalgae systems. It is used worldwide in microalgae cultivation in liquid media due to its easy accessibility, low cost, and ability to be ordered in desired volumes [12]. In addition, it has application areas, such as biomass production, biofuel production, and wastewater treatment [13,14]. Accordingly, bag systems are frequently used for commercial and academic purposes due to their rapid growth and high productivity, easy homogenization, cheapness, easy addition to the nutrient medium, ease of work, easy cleaning, easy detection, and the intervention of possible contamination [15]. Using heat-resistant plastic bags as containers during production is common in industrial mushroom production. Specifically, micro-holed plastic bags have become popular among *P. ostreatus* producers. However, bag systems have some disadvantages compared to conventional tray bioreactors due to their limited gas exchange, insufficient air circulation, inability to remove heat caused by microbial growth, and limited heat transfer [16]. Therefore, bigger-scale mushroom production facilities mostly prefer tray-type bioreactors, and in some cases, a mixed use of bag systems and tray reactors is chosen. Tray systems are used in defined spaces (rooms, tents, greenhouses, etc.) with climate control, and they require vast space and expensive equipment. However, according to tray bioreactors, bag systems provide an advantage when limited air circulation and heat exchange issues are fixed (Table 1).

Conclusively, as stated, bag systems are relatively sufficient for biomass production on a laboratory scale or on a large scale. However, their most important disadvantage is their insufficient air circulation, which reduces production efficiency in biomass growth. Their advantages, such as ventilation, mixing, and moisture retention, make them an efficient tool for solid-state bioreactor designs for controlled mass production. However, such systems lack air circulation, controlled sterility, and humidity control.

In this study, a modular bioreactor was developed to eliminate the weaknesses of bag bioreactors, especially for the three main properties (air circulation, controlled sterility, and humidity control). This bioreactor, named in the study as the modular chain bioreactor (MCB), contains candy-bag modules connected with chain connectors (CCs). The candy bags are pouches with both sides open. A cotton filter was installed on both parts of the bag for high air circulation, and both cotton filters were fixed with a cardboard cup. Finally, this module was duplicated and connected by CCs to form the MCB.

To examine the effect of the developed modular bioreactor on the production of fungi in solid-state cultures, biomass production was investigated using *G. lucidum*. In addition, the prevention of the spread of contamination, which is the main risk factor in mushroom cultivation and causes great losses, was examined.

## 2. Materials and Methods

### 2.1. Substrates and Microorganism

*G. lucidum*, one of the most studied species of white-rot Basidiomycetes, was used for the pilot tests of this study. The fungus was maintained on malt extract agar (MEA) slants at 4 °C. In a different ongoing study by the authors, inoculum production was optimized for pH, time, and an initial amount of inoculum. Accordingly, inoculum production was prepared using malt extract broth (MEB) (pH 5) by incubation at 28 °C and 200 rpm for 11 days. After incubation, the inoculum (Figure 1a) was used as 1 mL/5 g of dry substrate.

In order to see the effect of the system designed for biomass production, three candy bags were prepared for the MCB. One candy bag and one traditional bag were prepared as a control group. Both bags were filled with 200 g of zeolite support material and were enriched with a liquid solution containing mineral salts and nutrients (Table 2). Zeolite is an inert support material, consisting of SiO4 and/or AlO4, and is ideal for mycelium production in solid-state fermentation, thanks to the large voids in its structure [17]. When zeolite reaches the ideal humidity with a liquid solution containing nutrients, it has a structure that triggers mycelium production and keeps the pH value of the microorganism constant [18]. Therefore, this solution is also used as a moistening liquid to trigger microorganism growth.

### 2.2. Modular Chain Bioreactor (MCB) Prototype & Arduino Setup for Environmental Control

Solid-state bioreactors are an engineering and design issue for cultivating these organisms for research and industrial purposes. Although there are many different types of bioreactors, the main design principles of solid-state bioreactors focus on several points. These designs should be able to:Keep environmental conditions at optimum levels that are favored by the organisms (temperature, water activity, oxygen concentration),Carry on solutions for ventilation and mixing required by the production process,Block any organism from entering inside to prevent contamination and keep colonized organisms inside to expel any harmful effect that the organism can cause,Be produced by a durable and corrosion-resistant material, which should also not cause any toxic effect on colonized organisms-,Allow sampling and observation,Be suitable for any process required for solid-state fermentation (substrate preparation, inoculation, loading, unloading, sterilization),Be economic [19,20].

Traditionally used bag containers are temperature-resistant bags with a substrate added. The required air is supplied to the mouth of the bag with a cotton filter supported by a cardboard neck collar. Thus, while the containers can reach the air needed by the microorganisms, the substrate bed is protected from contamination. In order to enhance the properties of air circulation, controlled sterility, and humidity, a secondary cotton-filtered opening on the opposite side were examined first. It was observed that this “candy”-like bag container increased the colonization speed of the fungus (Figure 1). Enzymes are one of the most important products produced in solid-state fermentation. Microorganism growth in solid-state fermentation also affects enzyme production [21]. Culture conditions (pH, temperature, humidity, etc.) should be at optimum levels for the microbial used for rapid microbial growth [22]. However, one of the most fundamental limitations of enzyme production in solid-state fermentation is the measurement accuracy of the enzyme obtained. This may be due to the extraction method, the difference in the substrate used in the enzyme determination, and the fermentation parameters [23]. The developed candy-bag system provides better air circulation compared to traditional bag systems. The two-mouth candy-bag system proposed in the author’s ongoing solid-state fermentation and enzyme production study was compared with the traditional single-mouth bag system, and 1.8 times higher enzyme activity was observed in the candy-bag system compared to the bag system (data not shown).

This study used 45 × 50 cm polypropylene bags as candy-bag bioreactor systems for biomass production. Cotton was used as an air filter and cardboard cups (with the bottoms cut off) were used as scaffolding to keep the cotton stable.

However, fixed humidity control and problems in air circulation still occurred due to the crumpled structure of plastic bags and sterility limits. To solve these problems, the MCB was generated using separable, filtered, interlocking Chain Connectors (CCs) between the candy bags (Figure 2). It was designed in Fusion 360, and prototypes were developed by 3D printing. In order to test the design proposal, a system prototype was installed for 7 days and observed daily.

The proposed bioreactor aims to offer an improvement on the following qualities:Humidity-controlled environmentBetter air circulationModularity (each colonization unit can be separated from the chain without risking others with contamination)Sterile conditionsLow cost

Three heat-resistant colonization bags (candy bags) with identical substrates were linked for the testing process. A specially designed CC unit connected these bags in a chain. The main purpose of these connectors was to achieve better environmental control in the bags, while yielding the removal of desired bags without contaminating the system. An air blower with a 100 mm air outlet and a DC motor of about 100 W accommodated in the chain circulated the air through bags to ventilate the heating caused by the fermentation process. This motor blows 15 s of air per 3 min to circulate the air held inside the candy bags connected in chains through CCs. Meanwhile, a central humidifier mechanism feeds the system via silicon pipes connected to CCs with steam to keep the system in the range of 60–80% moisture, responsively to sensor readings. The relative humidity was adjusted simultaneously with the data collected from humidity sensors located within the CCs. In this study, Arduino Mega was used as a microcontroller to process the data gathered from sensors (DHT11) to maintain the relative humidity and air circulation at a desired level inside the bags by controlling air blowers and humidifier mechanisms, with the help of DC relays. The whole setup was constructed in a room with an air conditioner to cool the place to desired production temperatures between 25 and 28 °C. The temperature levels inside the bags were constantly monitored and recorded on an SD card loaded on the Arduino.

The humidifier mechanism consists of an air blower with a 75 mm air outlet and a DC motor of about 60 W, a sterilized humidifier with a 4 L water capacity, and pipes. The working principle of the humidifier is simply that an air blower is connected to the tank’s top level, forcing moisture to circulate inside the colonization bags. This process is controlled with the microcontroller. Figure 3 and Figure 4 illustrate a simplified block diagram of the electrical and mechanical system of the MCB and the humidifier.

The MCB system can be considered an example of lean entrepreneurship to prevent contamination with its modularity. The most important part of this system is the CC apparatus and the filters between this apparatus. Unlike traditional bag systems, the additional cost of filters is $1.80 per bag.

### 2.3. Fungal Biomass Production Experiments

The efficiency of the MCB prototype was examined by colonizing a pure culture of *G. lucidum* from the author’s previous research [24]. White-rot fungi, *G. lucidum*, with easy colonization and popular in academic research, was chosen as the control culture for the repetition of the tests. The zeolite was used as an inert support material to absorb the nutrients and minerals required for mycelium production and to be used at optimum pH levels [17,25]. Zeolite obtained from volcanic rocks is a mixed mineral-salt-medium solution as a nutrient broth, used as the substrate to achieve fast colonization and easy determination of the biomass amount. A pure culture of fungus was inoculated on the substrate for colonization. As can be expected from inorganic zeolite, its mass did not change during colonization, while fungi increased their mass by using nutrients. The dry mass obtained at the end of the incubation was proportional to the biomass of the fungus. The following equation was used in the calculations: m_(final dry weight)_ = m_(zeolite dry)_ + m_(mycelium biomass dry)_
(1)

### 2.4. Contamination Spread Pilot Experiment

In the MCB, a three-bag experiment was designed to measure the effect of the filter system and the contamination spread between the bags. The first and third bags were prepared under sterile conditions, and *G. lucidum* was inoculated. The second bag, which had filtered connections with other bags, was prepared under non-sterile conditions and was kept in the solid-state fermentation laboratory for half an hour before being included in the MCB. The bags were then connected to the MCB and supplied with humidity and air.

## 3. Results and Discussion

### 3.1. Contamination Control Experiments

In the three-bag experiment, the second bag in the middle was kept in a non-sterile area and added to the system to control the spread of contamination between modular systems. Three observations were made on the 3rd, 5th, and 7th days (Figure 5). Visual inspection shows apparent white hyphae (*G. lucidum*) colonizing the first and the third bags without any contamination; however, an unknown microorganism (orange-pink color) grew predominantly in the non-sterile bag (middle unit). The contamination in the second bag did not spread to other bags, indicating that the modularity of the filtration system and the developed MCB system were successful. If the filtration system had been inadequate and the contamination had spread to bags one and three, it would have also inhibited the mycelial production of *G. lucidum*. On the other hand, when Figure 5 is examined, the development of *G. lucidum* in bags one and three on the 5th and 7th days, and the contamination in the 2nd bag, can be clearly distinguished in terms of mycelium production and color. This shows that the filtration between modules successfully prevented the spread of contamination as intended (Figure 6).

As a result of the biomass production, 20.7 g and 11.3 g of mycelium production was observed in the bags (the 1st and 3rd bags, respectively). The 3rd bag did not obtain sufficient moisture due to water accumulation inside the humidifier’s silicon tube. A biomass production of 55.7 g was observed in the contaminated bag (Table 3).

### 3.2. Modular Chain Bioreactor System Results

Mycelium development as a result of biomass production in the MCB is given below (Figure 7). As a result of the experiment, it was seen that mycelium production was relatively rapid with a high humidity and high air supply. In environments where the amount of free water is high, the mycelium formation cannot increase, and the microorganism loses its viability. Because solid culture fermentation occurs in static conditions, if there is excess water in the environment, oxygen cannot be included in the substrate as there is no mixing as in submerged fermentation. Therefore, the dissolved oxygen contained in the excess water is consumed in a short time, and anaerobic conditions are formed. As a result, the diffusion of O_2_, which is needed for mycelium production, is prevented [21]. It should be taken into account in future studies that anaerobic conditions may have prevented aerobic contaminants. The main issue to be considered here is the O_2_-CO_2_ diffusion between the biomass and the substrate. In the presence of free water, this diffusion is restricted and inhibits aerobic microbial growth. In this case, the mycelium cannot reach the oxygen it needs, and microbial growth remains weak. However, because the moisture in the 3rd bag was too high, free water accumulation occurred in the environment. While the amount of free water was 0 mL in the inoculum fluid at the beginning of the experiment, approximately 25 mL of water accumulation was measured at the end. In environments where the amount of free water is high, the mycelium cannot grow, and the microorganism loses its viability. Therefore, the moisture supply to the environment during fermentation must be optimized so that it is neither too low nor too high to cause free water accumulation. Compared to conventional bags, a more successful result was obtained with the mycelial biomass developed in the MCB candy bag. Mycelial production in the moisture- and air-fed MCB was three times higher than that in the candy bag and mycelial biomass formation was 3.5 times higher than in conventional bag systems (Table 4).

According to the results, the conditions provided for mycelium production during the 7 days showed a positive effect. The provided air supply and humidification were shown to provide early microbial activation. In addition, the fact that mixing can be achieved manually accelerates the growth of microorganisms that are activated early. However, during the experiment, it was observed that water accumulation occurred in the second bag. This negative situation can prevent mycelial production in solid-state fermentation. For this reason, as a result of the obtained biomass, the production in the 2^nd^ bag was lower than in the other bags (Figure 8).

Another outcome of the experiments can be derived as follows: the design of the MCB, yields the potential to grow different kinds of organisms without affecting or contaminating each other directly. This way, different stages of mycelium development can be kept together, and the faster colonies can be harvested easily. Another outcome of the MCB would be the ability to multi-use. For example, while one bag houses spore-producing fruiting of medicinal *G. lucidum*, the neighbor bag can be a resident for *T. versicolor*, produced for enzyme subtraction. This chain can be continued for any use requiring similar environmental conditions, ranging from commercial to personal use.

## 4. Conclusions

In this study, a modular bioreactor chain system was produced to eliminate the disadvantages of bag systems frequently used in solid culture fermentation. In a modular system, parameters, such as environment control, sensor systems, and modular connectivity, were discussed, and a particular connection system named CC was developed to connect the modules. The MCB is constituted of connected modules and is a modular chain bioreactor prototyped to test the efficiency of the setup.

Within the scope of the study, the prototype produced with three bag modules with a two-filtered opening contained equivalent solid culture samples connected by CCs. There was a humidity/temperature sensor and a hose-pipe connection in the CCs. In addition, air blowers were added to provide air and humidity flow to the chamber, and these systems were connected with Arduino to provide environmental control. A double-sided filtration system was developed to minimize the risk of contamination between the CCs and bags to provide humidity control. Typical bag containers mostly have one hole for air circulation, and in the case of mass production scenarios, each bag is vulnerable during operation times. However, the double-sided filtration system fixes this vulnerability by connecting all openings in one filtered line for airflow. It allows the chosen unit to be removed without disturbing the rest of the chain.

Biomass production and contamination spread experiments were performed to show that the system works effectively in this study. In the pilot contamination experiments, the second module was left in a non-sterile environment, and the development of all three samples was observed. As a result of the 7-day mycelium-development observation in all modules, when contamination was observed in the second module as expected, the absence of contamination in other modules supported the system’s efficiency.

According to the biomass production trial experiments, it has been predicted that the ventilation and humidification system designed in the MCB could be more effective than traditional bag systems. Furthermore, the system’s modularity offers the potential to work with different microorganisms simultaneously, the option of harvesting at different times in the same production, and the loss of a single bag rather than the whole production in case of possible contamination. The results of the experiments and observations through the process revealed these advantages of the current MCB prototype in solid-state fermentation:High ventilation capacityMoisture level controlEasy mixingEfficient contamination controlPermitting cultivation and harvesting without risking the rest of the bags through modularityCultivability of different microorganisms for different purposesObservability of mycelial growth through the processLow cost

However, there are still some issues with the system that need improvement. The accumulation of water in the feed hoses during humidification in the experiments and the flow of this accumulated water to the environment cause the risk of preventing mycelium production. The length of the silicon tubing should be adjusted to prevent water accumulation in further tests. In addition, the air blowers’ response sequences and the optimization of the sensor reading times are required. The air blower’s power capacity should be adjusted according to the filter’s number and resistance to the direct blow of the wind. In addition, another factor to be considered in future studies is the determination of how long the prepared system can be used by calculating the efficiency in terms of gram biomass/gram substrate. This study used solid support materials to compare the MCB system and traditional bag systems. However, using a solid substrate instead of a solid support material will increase the yield for more extended use. Finally, the MCB needs more space than the standard bag containers, trained personnel, and the risk of damage to sensitive system elements due to moisture and disinfection chemicals.

## Figures and Tables

**Figure 1 biomimetics-07-00179-f001:**
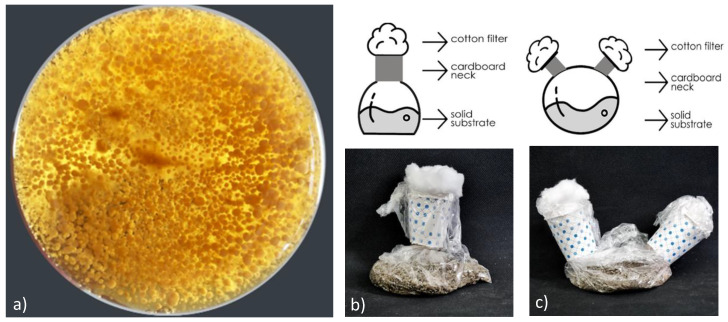
(**a**) Pellets (bottom the flask); (**b**) cotton-filtered bag container; (**c**) Candy-bag bioreactor (right).

**Figure 2 biomimetics-07-00179-f002:**
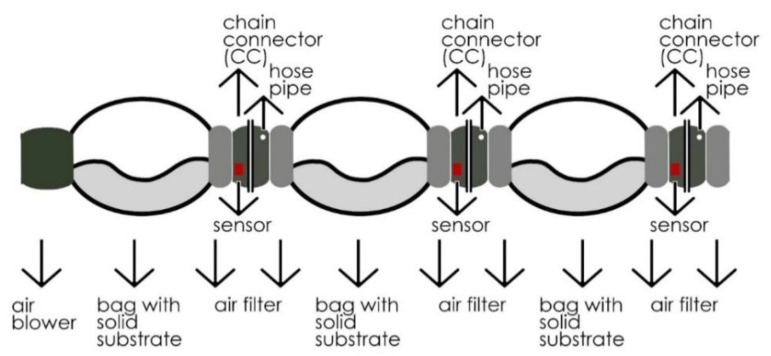
MCB—3 Unit chain scheme.

**Figure 3 biomimetics-07-00179-f003:**
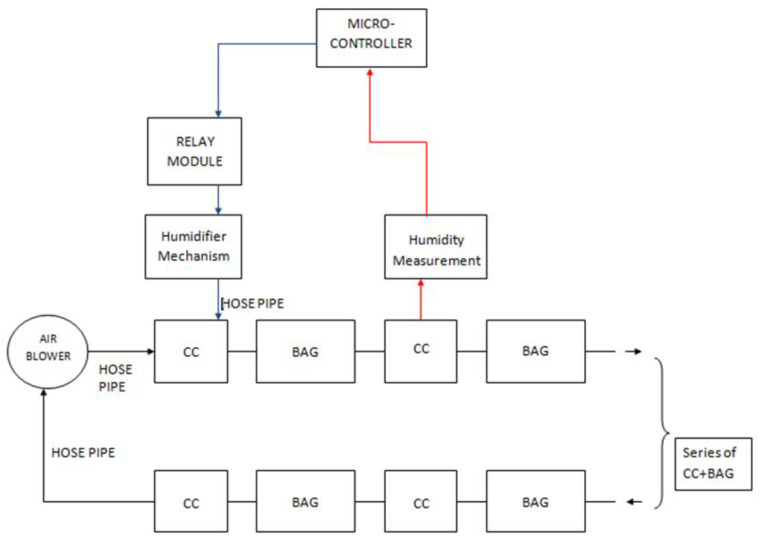
Simplified block diagram of MCB.

**Figure 4 biomimetics-07-00179-f004:**
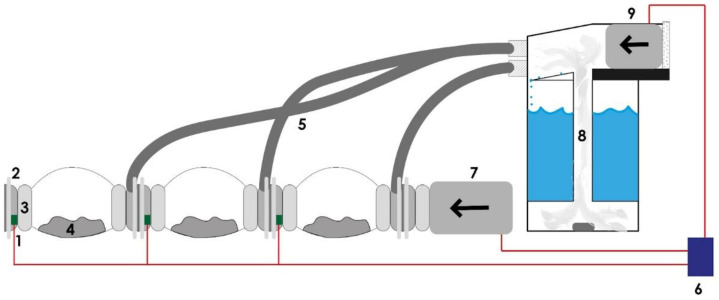
The simplified electrical and mechanical system of the MCB and the humidifier: (1) humidity/temperature sensor, (2) CC, (3) air filter, (4) solid substrate, (5) hose pipe, (6) Arduino, (7) 100 W air blower, (8) vapor chamber, (9) 50 W air blower.

**Figure 5 biomimetics-07-00179-f005:**
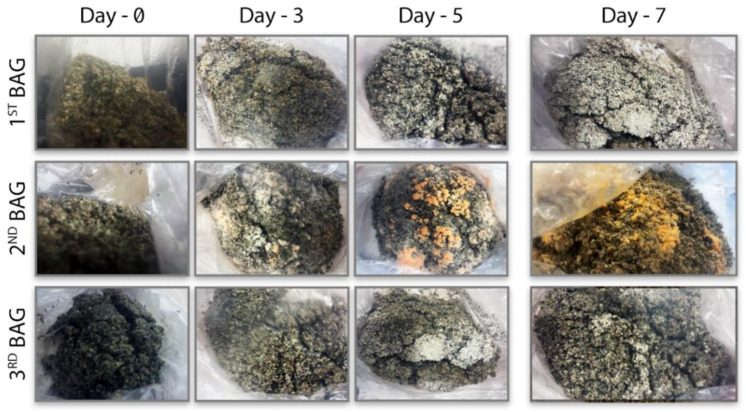
MCB contamination control experiments.

**Figure 6 biomimetics-07-00179-f006:**
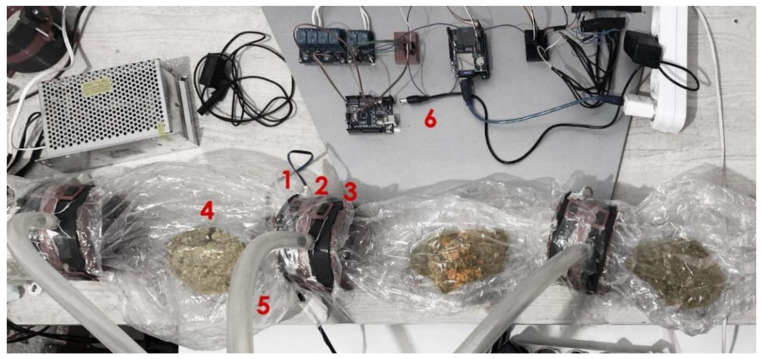
MCB contamination control experiment setup: 1, humidity/temperature sensor; 2, CC; 3, air filter; 4, solid substrate; 5, hose pipe; 6, Arduino).

**Figure 7 biomimetics-07-00179-f007:**
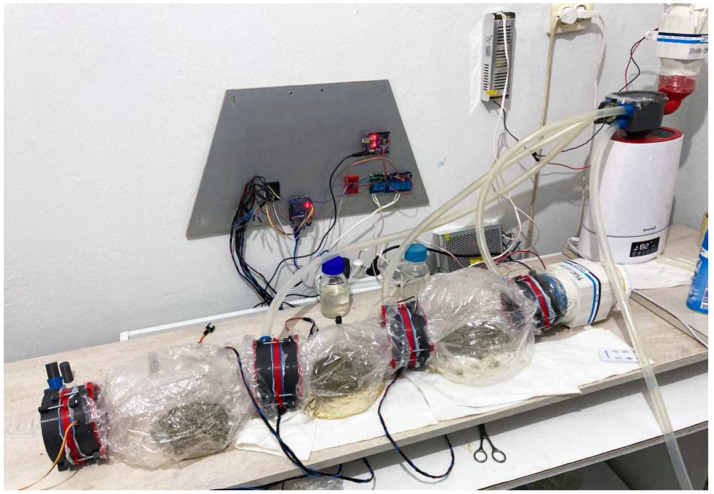
MCB setup for biomass experiments.

**Figure 8 biomimetics-07-00179-f008:**
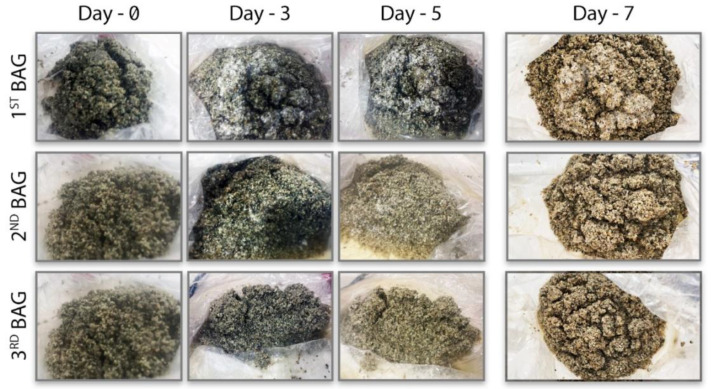
MCB biomass experiment observations.

**Table 1 biomimetics-07-00179-t001:** Comparison of bag system and tray bioreactor.

Bag System	Tray Bioreactor
Easy to sterilize	Sterilization is more difficult than the bag system
Can be mixed during fermentation	Cannot be mixed effectively
Cheap	More expensive than bag system
Limited air circulation	Efficient air circulation
Limited heat exchange	Efficient heat exchange
Hard to remove microbial heat in fermentation	Microbial heat formed during fermentation can be removed
Low area requirement	Wide area requirement
Microbial growth is controllable	Microbial growth is uncontrollable

**Table 2 biomimetics-07-00179-t002:** Nutrient-mineral salt solution.

Composition	Formula	Amount (g/L)
Glucose	C_6_H_12_O_6_	30
Peptone		2
Ammonium nitrate	NH_4_NO_3_	2
Trisodium citrate dihydrate	C_6_H_5_Na_3_O_7_·2H_2_O	2.5
Potassium dihydrogen phosphate	KH_2_PO_4_	5
Magnesium sulfate heptahydrate	MgSO_4_·7H_2_O	0.2
Calcium chloride dihydrate	CaCl_2_·2H_2_O	0.1
Di ammonium hydrogen phosphate	(NH_4_)_2_HPO_4_	4

**Table 3 biomimetics-07-00179-t003:** Contaminant control experiment assay.

Bag	Final Weight (g)	Final Dry Weight (g)	Zeolite (g)	Biomass (g)
1	264.4	220.7	200	20.7
2	285.2	255.7	200	55.7
3	255.4	211.3	200	11.3

**Table 4 biomimetics-07-00179-t004:** MCB experiment biomass assay.

Bag	Final Weight	Final Dry Weight	Zeolite	Biomass
1	263.88	221.88	200	21.88
2	251.1	217.4	200	17.4
3	271.5	220.3	200	20.3
Control candy bag	261.3	209.7	200	9.7
Control traditional bag	248.2	206.1	200	6.1

## Data Availability

Data are available upon request.

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
