# Peer review of "A Modular Chain Bioreactor Design for Fungal Productions"

_biomimetics, 2022, doi:10.3390/biomimetics7040179_

Round 1
Reviewer 1 Report
The work presented is interesting and ingenious and fits within the scope of the journal. The proposed system of fermentation in solid medium based on the use of bags that eliminates several problems that had been presented in other works, since the articulation of a modular system makes it possible to eliminate those bags that could have been contaminated. In this sense, I consider that it is a system with potential to be used in fermentation systems in solid medium. However, there are several considerations that must be taken into account in the presented work.
1. In line 138-139 it says
“The two-mouth Candy-bag system proposed in the author's ongoing solid-state fermentation and enzyme production study was compared with the traditional single-mouth bag system, and 1.8 times higher enzyme activity was observed in the Candy-bag system compared to the bag system (data not shown).
In the scientific literature it has been published that the measurement of enzymatic activity in solid media is problematic because the medium is embedded in the support, which makes recovery difficult. This explanation should be duly referenced in the document. On the other hand, if it has been determined, the reference should also be present.
2. In line 215-216 it says
“Three observations were made on the 1st, 4th and 7th days (Fig 5). Visual inspection shows clear white hyphae (G. lucidum)…”
The days cited in the text do not correspond to the data in figure 5, the data in the figure and in the text must be consistent.
3. In line 2015 it says
“Visual inspection shows clear white hyphae (G. lucidum) colonizing on the first and the third bags without any contamination;
This part of the work is the one that in my opinion constitutes a strong criticism of the experimental design. The assertion in the text that visual inspection can be an accurate methodology to ensure the absence of contamination. The absence of contamination in the bags should have been determined at least by identification of the morphology of the hyphae and spores through microscopy images, which is absent in the document. Additionally, the techniques that would ensure that there is only one microorganism in a culture would require a molecular characterization of the sample at the beginning and after the culture. The reasons for doing these tests are because the non-sterile conditions of bag 2 resulted in the system becoming contaminated, and perhaps it was only a matter of time before bags 1 and 3 began to show the presence of the contaminating microorganism. The time in which the presence of contamination was evaluated is not a sufficient criterion, since competition phenomena occur in an inoculated medium that can have bacteriostatic effects on the organisms that try to colonize the medium, which paints a picture in which there are presence of the contaminant, but it is not visible. In the work it is assumed that there is no contamination, but visual inspection is not an acceptable scientific criterion.
4. In line 233 it says
“As a result of the experiment, it is seen that mycelium growth is quite rapid with high humidity and high air supply”
In this part of the document, an expression is used again that should be supported by scientific terminology, adjusted to the parameter that in Microbial Physiology is used to refer to growth rate, it is the specific growth rate (m), published in an enormous amount of works. This way to refer microbial growth would only be acceptable by referring to a figure in the document where the specific growth rate is shown. But the figure is not there because the growth was not measured.
5. In line 234 it says
“However, since the amount of moisture in the 3rd bag was too high, free water accumulation occurred in the environment”
The document speaks of water accumulation, however, it is never quantified, and it should have been quantified, since later, the presence of water is related to an effect on growth.
6. In line 235 it says
"In environments where the amount of free water is high, the mycelium cannot grow and the microorganism loses its viability."
The loss of viability must be adjusted to terms such as decay rate in a curve that represents the evolution of the biomass over time compared to optimal conditions (where the amount of water is controlled). In this sense, it would have been useful to design an experiment to determine the amount of water that condenses, and in any case, it should be in numerical terms in order to establish a comparative criterion that could eventually be optimized.
On the other hand, although it is not stated in the document, it is assumed that the zeolite is capable of homogeneously distributing the nutrients of the medium and the minerals as well as maintaining an optimal pH, this should be discussed and referenced, in any case, the media of solid fermentations are usually intrinsically heterogeneous, which could explain the differences in biomass, in this sense in the document they are only justified by the differential accumulation of water, that should be adequately justified, referenced, reasoned.
7. In reference to the second part of the work, I consider that although they are related experiments, different data are presented, and therefore I understand that they should be presented separately.
On the other hand, in the second part of the work a series of figures are shown, in which I understand that it is intended to show the utility as an interior lighting system, however, it is convenient to include several aspects to adjust to a more scientific theoretical framework.
a) The experimental strategy should include a mathematical treatment that would allow correlating the amount of light that is emitted as a function of the biomass that is produced. Since this methodology would make it possible to determine the optimization of nutrients, as well as the resources destined for air circulation and humidity maintenance, however, this part of the work is absent from the document.
b) On the other hand, I consider it pertinent to indicate that the intensity of the light generated by the mycelium of the fungus should be quantified in lumens, since otherwise it is not possible to determine the usefulness of the emitted light. In this sense, the photographs provided are not of a scientific nature but are illustrative.
Reviewer 2 Report
Title: please revise
Abstract: remove 'Thanks', its not a proper wording
Figure 8: please zoom in or provide microscopic view of the mycelium-substrate observation, and discuss whether the mycelium has fully colonized the substrate
Provide cost efficiency of using MCB compared to standard bag system
The inoculum was prepared using malt extract broth (MEB) (pH 5) at 200 rpm, 28°C for 11 days. The mycelium was inoculated as 1 ml/5 g of dry substrate. (Elaborate this part with pictures, what was the morphology of inoculum before inoculation?) Why 11 days? have u optimized the condition.
Provide the source of T. versicolor and bioluminescent P. stipticus, any picture of P. stipticus?
If light generation is the main idea, have u calculated the light intensity produced by P. stipticus?
Round 2
Reviewer 1 Report
Once the document has been reviewed, a clear improvement is perceived, since several recommendations were introduced that were suggested in the first review, however, there are basically 3 observations that need to be emphasized.
4. In line 233 it says
“As a result of the experiment, it is seen that mycelium growth is quite rapid with high humidity and high air supply”
In this part of the document, an expression is used again that should be supported by scientific terminology, adjusted to the parameter that in Microbial Physiology is used to refer to growth rate, it is the specific growth rate (m), published in an enormous amount of works. This way to refer microbial growth would only be acceptable by referring to a figure in the document where the specific growth rate is shown. But the figure is not there because the growth was not measured.
In their answer they indicate that the calculation of the specific growth rate is problematic in cultures in solid medium. In this sense, it is necessary to mention that it is possible to directly monitor the biomass that is formed in these conditions, for example, through the production of CO2 that is generated as a consequence of the consumption of substrate, or through the consumption of oxygen. But regardless of this, you state that your protocol included the measurement of dry weight to analyze the parameters of growth kinetics, however, in the text, you only include a mathematical expression of how you made the calculation, but the data does not appear in the document, neither in the form of a table nor a figure and therefore the maximum biomass, nor the specific growth rate, which are fundamental variables, do not appear. In my opinion this is clearly insufficient.
Elsewhere in your answers it says
“However, since solid culture fermentation takes place in static conditions, if there is excess water in the environment, oxygen cannot be included in the system as there is no mixing as in submerged fermentation, and the dissolved oxygen contained in the excess water is consumed in a short time and anaerobic conditions are formed.”
This statement is interesting in the sense that it is assumed that anaerobic conditions are reached, in this sense the answer could be related to another of the comments, where they refer to the fact that the filtration process in the reactor prevented contamination. In this sense, it could be reasoned that the absence of polluting organisms is due not to the filtering system but to the anaerobic conditions. On the other hand, for an organism such as G. Lucidum they should have affected growth causing a fungistatic effect, which should be detectable in the biomass dry weight values. However, these data are absent from the document.
Finally, in reference to the last part of the work, the fact of adding the clarification to the document that it is a design proposal is not a significant enough element, from my point of view this part of the work still lacks an analysis mathematical model that correlates the intensity of light emitted with the biomass produced. Therefore, the fact that it is illustrative and not scientific information has not changed.
Round 3
Reviewer 1 Report
4. In line 233 it says
“As a result of the experiment, it is seen that mycelium growth is quite rapid with high humidity and high air supply”
In this part of the document, an expression is used again that should be supported by scientific
terminology, adjusted to the parameter that in Microbial Physiology is used to refer to growth rate, it is the specific growth rate (m), published in an enormous amount of works. This way to refer microbial growth would only be acceptable by referring to a figure in the document where the specific growth rate is shown. But the figure is not there because the growth was not measured.
In their answer they indicate that the calculation of the specific growth rate is problematic in cultures in solid medium. In this sense, it is necessary to mention that it is possible to directly monitor the biomass that is formed in these conditions, for example, through the production of CO2 that is generated as a consequence of the consumption of substrate, or through the consumption of oxygen. But regardless of this, you state that your protocol included the measurement of dry weight to analyze the parameters of growth kinetics, however, in the text, you only include a mathematical expression of how you made the calculation, but the data does not appear in the document, neither in the form of a table nor a figure and therefore the maximum biomass, nor the specific growth rate, which are fundamental variables, do not appear. In my opinion this is clearly insufficient.
As you mentioned, results can also be obtained by measuring CO2 production and/or O2 consumption. Since zeolite, which is used as a support material in solid culture, cannot be consumed by microorganisms, it was decided that biomass determination by dry weight measurement was appropriate in the study. In order to interpret the success of the presented MCB system, two control groups, one traditional bag and one candy-bag, were observed. The control groups used in the experiment were kept in the same environmental conditions in the MCB system and had the same nutrient medium. Results from the MCB system and control groups are presented as a table in the manuscript. In the table final weight, post-drying weight and initial support material weight are included, based on these, biomass measurement was made.
In reference to your response, and considering the logistical complications of gas measurement, which involves coupling a detection mechanism to your fermentation system in a solid medium, it is pertinent to indicate that it would be a valid measurement in total terms. However, it is not an average that allows evaluating the evolution of biomass production and obtaining kinetic parameters such as the specific growth rate and other interesting parameters such as the duration of the lag phase, which are important variables that must be considered for purposes of estimation. optimize the light production process, which would be desirable in future works. Another element that, from my point of view, must be considered in this study refers to the yield (gram biomase/gram substrate), which is an indication of the amount of culture medium that must be used to calculate the time for which the fungus can emit light. (his work does not have it and it would be desirable that it be considered in future studies)
“However, since solid culture fermentation takes place in static conditions, if there is excess water in the environment, oxygen cannot be included in the system as there is no mixing as in submerged fermentation, and the dissolved oxygen contained in the excess water is consumed in a short time and anaerobic conditions are formed.” This statement is interesting in the sense that it is assumed that anaerobic conditions are reached, in this sense the answer could be related to another of the comments, where they refer to the fact that the filtration process in the reactor prevented contamination. In this sense, it could be reasoned that the absence of polluting organisms is due not to the filtering system but to the anaerobic conditions. On the other hand, for an organism such as G. Lucidum they should have affected growth causing a fungistatic effect, which should be detectable in the biomass dry weight values. However, these data are absent from the document.
Due to the accumulation of free water, good aeration did not occur in the lower parts of the solid medium, so mycelial growth was not observed in this part, while mycelial growth of Ganoderma lucidum was observed in the upper part. On the other hand, the contaminant-specific colonial morphology seen in the other bag was not observed in this bag. Therefore, no relationship was established between the free water accumulation and the contamination.
In reference to your answer, in my opinion you are not considering the postulates that I described to you in my initial planning, in your confirmatory test of the presence of polluting organisms, I indicate that there is the possibility that an anaerobic context is generated and therefore the The experimental strategy that must be articulated to detect this type of organisms should consider anaerobic conditions and the experimental model of the Petri dishes that were used do not give oxygen restriction conditions. Therefore, they do not have the certainty of assuming that there is no presence of contaminating organisms.
Finally, in reference to the last part of the work, the fact of adding the clarification to the document that it is a design proposal is not a significant enough element, from my point of view this part of the work still lacks an analysis mathematical model that correlates the intensity of light emitted with the biomass produced. Therefore, the fact that it is illustrative and not scientific information has not changada.
In line with the referee's suggestion, the only thing that can be done in this regard will be a detailed study focusing on the relationship between light amount and biomass in the future. Because it is not possible to test this design proposal in the real environment yet, it has not been possible to give a mathematical analysis model at this stage.
Regarding the correlation between the amount of mycelium with bioluminescent capacity, I understand that it is necessary to establish this correlation since the purpose of this fermentative mechanism would be to illuminate rooms of different sizes. In this sense, I consider it pertinent to correlate the size of the space to be illuminated with the lighting capacity of the fermentation system.
From a strictly scientific point of view, the entire final part of the document that refers to the illumination of the fungus subtracts scientific potential from the work and does not allow it to be accepted, the work contains novel elements in terms of fermentation and its potential use in other biotech applications, but I think the lighting part should be removed from the document.
